

# Integrated bioinformatics analysis of the NEDD4 family reveals a prognostic value of *NEDD4L* in clear-cell renal cell cancer

Hui Zhao[1,2,*]   Junjun Zhang[3,*]   Xiaoliang Fu[4]   Dongdong Mao[1]   Xuesen Qi[1]
Shuai Liang[1]   Gang Meng[1]   Zewen Song[3]   Ru Yang[5]   Zhenni Guo[6]
Binghua Tong[6]   Meiqing Sun[6]   Baile Zuo[7]   Guoyin Li[6,8]

[1] Department of Urology, Affiliated Hospital of Weifang Medical University, Weifang, China
[2] Department of Urology, China Rehabilitation Research Centre, Rehabilitation School of Capital Medical University, Beijing, China
[3] Department of Oncology, The Third Xiangya Hospital of Central South University, Changsha, China
[4] Department of Urology, The Second Affiliated Hospital of Air Force Medical University, Xian, China
[5] Henan Key Laboratory of Neurorestoratology, The First Affliated Hospital of Xinxiang Medical University, Weihui, China
[6] College of Life Science and Agronomy, Zhoukou Normal University, Zhoukou, China
[7] Tumor Molecular Immunology and Immunotherapy Laboratory, School of Laboratory Medicine, Xinxiang Medical University, Xinxiang, China
[8] Academy of Medical Science, Zhengzhou University, Zhengzhou, China
[*] These authors contributed equally to this work.

Corresponding authors
Baile Zuo, zuobaile@xxmu.edu.cn
Guoyin Li, 20181004@zknu.edu.cn,
guoyin0604@163.com

## ABSTRACT

The members of the Nedd4-like E3 family participate in various biological processes. However, their role in clear cell renal cell carcinoma (ccRCC) is not clear. This study systematically analyzed the Nedd4-like E3 family members in ccRCC data sets from multiple publicly available databases. NEDD4L was identified as the only NEDD4 family member differentially expressed in ccRCC compared with normal samples. Bioinformatics tools were used to characterize the function of NEDD4L in ccRCC. It indicated that NEDD4L might regulate cellular energy metabolism by co-expression analysis, and subsequent gene ontology (GO) and Kyoto Encyclopedia of Genes and Genomes (KEGG) analysis. A prognostic model developed by the LASSO Cox regression method showed a relatively good predictive value in training and testing data sets. The result revealed that NEDD4L was associated with biosynthesis and metabolism of ccRCC. Since NEDD4L is downregulated and dysregulation of metabolism is involved in tumor progression, NEDD4L might be a potential therapeutic target in ccRCC.

## INTRODUCTION

Renal cell carcinoma (RCC) is the third most common malignant tumor in the urinary system (*Bray et al., 2018*). Recently the incidence of RCC has been increasing, and approximately 403,262 new cases and 175,098 kidney cancer-related deaths occurred worldwide in 2018 (*Siegel, Miller & Jemal, 2019*). Clear-cell renal cell carcinoma (ccRCC) is the most common pathological type and accounts for about two-thirds of all RCCs
(*Kroeger et al., 2013*). Nearly 30–40% of patients with ccRCC will develop metastases (*Ghatalia et al., 2019*; *Pierorazio et al., 2016*). The Von Hippel-Lindau/hypoxia-inducible factor (VHL/HIF) axis, the major carcinogenic pathway in ccRCC, promotes angiogenesis, cell growth and glycolysis by activating vascular endothelial growth factor A (VEGFA), transforming growth factor alpha and beta and platelet derived growth factor (*Seles et al., 2020*; *Sanchez-Gastaldo et al., 2017*; *Maxwell et al., 1999*). Compared to chemotherapy and radiotherapy, surgical resection of early-stage renal cancer is more effective (*Chow et al., 2010*). However, 20–40% of cases will relapse after surgery (*Janzen et al., 2003*). Accordingly, it is necessary to find biomarkers that can be effectively used to monitor RCC progression and evaluate its prognosis. Research shows that E3 ubiquitin ligases take part in various biological processes involved in the pathogenesis of carcinoma which could be considered as potential biomarker candidates (*Wang et al., 2017*).

The Nedd4-like E3 family mediates ubiquitination and proteasomal degradation of its substrate proteins as well as receptor-mediated endocytosis. In mammals, there are nine members: NEDD4, NEDD4L, ITCH, SMURF1, SMURF2, WWP1, WWP2, NEDL1 and NEDL2 (*Wang et al., 2020*). The Nedd4-like E3 family members have similar domains, including an N terminal C2 calcium-dependent phospholipid binding domain, 3–4 WW protein-protein interaction domains, and a C-terminal catalytic HECT ubiquitin ligase domain (*Boase & Kumar, 2015*). The Nedd4-like E3 family plays an important role in various biological processes. In recent years, the role of the NEDD4 family in urinary system tumors has attracted much attention. In bladder cancer, NEDD4 exerts its oncogenic function by regulating phosphatase and tensin homolog (PTEN) and notch receptor 1 (NOTCH1) expression (*Wen et al., 2017*). In prostate cancer, NEED4 exerts its oncogenic function by downregulating PTEN and androgen receptor (AR) (*Li et al., 2015*; *Li et al., 2008a*; *Li et al., 2008b*). NEDD4L is down-regulated in prostate cancer, and may exert an antitumor function by regulating TGFβ1 signaling (*Li et al., 2015*; *Li et al., 2008a*; *Li et al., 2008b*). However, research on the function of NEDD4 family members in ccRCC is lacking, and more attention needs to be paid to this topic.

The expression level of *NEDD4L* has been found significantly decreased in various tumors (*Tanksley, Chen & Coffey, 2013*; *Yang et al., 2015*; *Hu et al., 2009*; *Gao et al., 2012*; *Zhao et al., 2015*). It is reported that NEDD4L acts a pivotal part in the prognosis of various malignant carcinomas (*Jiang et al., 2019*). Also, NEDD4L has been shown to be involved in the regulation of certain major signaling pathways in carcinoma, including the WNT, and TGF-β signaling pathways (*Zou, Levy-Cohen & Blank, 2015*). In addition, NEDD4L has been shown to act as a cancer suppressor by inhibiting canonical WNT signaling in colorectal cancer (*Tanksley, Chen & Coffey, 2013*). Additionally, NEDD4L was found to suppress tumor metastasis by inhibiting the activity of TGF-β signaling pathway (*Kuratomi et al., 2005*). NEDD4L can bind to TGFBRI via SMAD7 and induce its degradation through ubiquitin proteasome pathway, thereby preventing the activation of R-SMADs. Moreover, NEDD4L can bind to SMAD2 and SMAD3 in a ligand-dependent manner and induce the degradation of SMAD2 (*Kuratomi et al., 2005*). However, NEDD4L was confirmed to have distinct functions in different carcinomas. Furthermore, NEDD4L was found to exert its pro-oncogenic function in gallbladder cancer, where it promotes invasion by

regulating transcription of the matrix metallopeptidase 1 and 13 genes (*Takeuchi et al., 2011*). However, the function of NEDD4L in ccRCC remains unknown.

Fatty acids metabolism exerts new functions in cancer, such as protecting cells from lipotoxicity, facilitating cell migration and drug resistance by altering membrane fluidity, and meeting high energy demands of metastatic cells through fatty acid β-oxidation (FAO)Tumor progression depends on the reprogramming of cell metabolism (*Pavlova & Thompson, 2016*). In recent years, elevated lipid synthesis is considered to be an important metabolic abnormality for tumorigenesis (*Swierczynski, Hebanowska & Sledzinski, 2014*). The biosynthesis of fatty acids is increased, and the metabolism of fatty acids exerts new functions in cancer, such as protecting cells from lipotoxicity, facilitating cell migration and drug resistance by altering membrane fluidity, and meeting high energy demands of metastatic cells through fatty acid β-oxidation (FAO) (*Chen & Huang, 2019*). The rapid proliferation of tumor cells also increases their demand for amino acids. In addition to being used for protein synthesis, amino acids can also be used as metabolites and regulators to support the growth of tumor cells (*Li & Zhang, 2016*).

There is not a lot of literature on NEDD4L and cell metabolism and these topics need more attention from researchers. In this paper, we integrated data from multiple public databases like the Cancer Genome Atlas (TCGA), Oncomine and gene expression database (GEO) to methodically study the expression profile, prognostic significance and role of NEDD4L in ccRCC. The results indicate that NEDD4L is markedly downregulated in ccRCC, which is related to poor prognosis. Enrichment analysis indicates that NEDD4L may regulate biosynthesis of unsaturated fatty acids and fatty acids metabolism. These findings reveal for the first time the effect of NEDD4L in the prognosis of ccRCC and the potential involvement of this protein in the regulation of fatty acid synthesis and metabolism.

## MATERIALS AND METHODS

### Data acquisition and processing

The kidney clear cell Carcinoma (KIRC) project of the TCGA (TCGA_KIRC) data set and GEO data sets (GSE40435 and GSE53757) were acquired and processed using the method described in our previous study (*Zhang et al., 2020*). Briefly, normalized RNA-seq data (607 cases, HTSeq-FPKM), phenotype information (985 cases), and survival data (979 cases) of the TCGA_KIRC were downloaded from the GDC hub of UCSC xena website (http://xena.ucsc.edu/public). Gene expression data of the GSE40435 and GSE53757 data sets (series matrix file) was downloaded from the Gene Expression Omnibus (GEO) database through the GEOquery package in the R software (version 3.6.2). After data processing, 526 tumor samples with survival data of the TCGA_KIRC data set, 202 samples of the GSE40435, and 144 samples of the GSE53757 were used for further analysis. All data for this study were acquired from public databases and did not require approval from the Ethics Committee.

## Online database analysis

The transcriptional level of NEDD4L in renal cell carcinoma was verified in the Oncomine database (https://www.oncomine.org/resource/login.html). The expression data of 5 groups of renal cell carcinoma and normal tissues (transformed by log2) were retrieved and compared statistically. The default threshold was as follows: P $<1E^{-4}$, multiple change >2, and the top 10% of genes. The staining intensity of NEDD4L in RCC samples and normal kidneys were downloaded from the Human Protein Atlas database (https://www.proteinatlas.org/).

## Functional analysis and enrichment analysis

Gene ontology (GO) and Kyoto Encyclopedia of Genes and Genomes (KEGG) enrichment analyses were carried out using the clusterProfiler package in R version 3.6.2 (*Hsing et al., 2020*). Gene set enrichment analysis (GSEA) was used to investigate pathways enriched in the high- and low-risk subgroups. C2.cp.k*egg.v7.1.symbols.gmt* was chosen as the gene set database. These pathways were considered to be significantly enriched when the following criteria were met: nominal *p*-value < 0.05, false discovery rate *q*-value < 0.25, and absolute normalized enrichment score >1.

## Development of the prognostic model

The least absolute shrinkage and selection operator (LASSO) Cox regression analysis was performed using the glmnet package in R. The analysis generated key gene signatures, and their corresponding coefficients were obtained by multi-variate cox analysis. A new score was calculated by multiplying the gene expression value of each gene and their corresponding coeffecient, as follows:

Score = − 0.06919*SOWAHB − 0.18041*WDR72 − 0.22124*EPB41L4A-DT − 0.07547*C4orf19 − 0.22609*CDKL2 − 0.05303*IRF6 − 0.16642*DHRS7 − 0.31446*NUPR2.

The risk score was calculated by subtracting the minimum score of the cohort from this score, and dividing the difference by the absolute value of the maximum score of the cohort, namely riskscore = (Score-min(Score))/abs(max(Score)).

## Development and evaluation of the nomogram

The independent risk factors determined by multivariate Cox regression analysis were chosen to develop a nomogram for prediction of total overall survival (OS) probability. The consistency index (C-index) was used to evaluate the consistency between the occurrence frequency of the actual results and the prediction probability of the model. The nomogram was generated using the rms package in R software (*Li et al., 2019a*; *Li et al., 2019b*).

## Statistical analysis

The data collected were analyzed by default as described using web resources. The rest of the data was analyzed using R software (version 3.6.2). The patients were divided into two subgroups by the median value of gene expression or risk score. Briefly, the correlation between the expression of NEDD4L and the clinicopathological features of ccRCC patients was examined by chi-square test with SPSS software (IBM Corp., Armonk, NY, USA). The

expression of NEDD4L and other genes was analyzed by Spearman correlation analysis with R software (version 3.6.2). Students' $t$-test was used to compare the level of gene expression. Univariate/multivariate Cox regression were performed by using the 'survminer' package in R. Kaplan–Meier survival analysis was used to compare the OS, relapse-free survival (RFS), disease specific survival (DSS) and progression free interval (PFI) of high- and low-CIFI groups Time-dependent receiver operator characteristic (ROC) analyses and subsequent calculation of the area under the curve (AUC) (*Le, 2019*; *Le et al., 2019*) were performed using the 'timeROC' package in R. Packages in R used for data analysis and graph plotting included ggstatsplot, ggplot2, ggpubr, limma, survminer, survival, tidyverse, dplyr, and plyr. A value of $p < 0.05$ was considered to be statistically significant (*, $p < 0.05$; **, $p < 0.01$; ***, $p < 0.001$; ****, $p < 0.0001$).

# RESULTS

## Patient characteristics

The data were collected from the TCGA_KIRC data set, including 72 normal subjects and 530 patients with primary tumors with clinical and gene expression data. The 526 patients with survival information were used for follow-up analysis (Table S1). There were 183 females (34.79%) and 343 males (65.21%). The onset age ranged from 26 to 88 years old, and the median age was 60 years old, 245 (46.58%) cases were younger than 60 years, and 281 (53.42%) cases were older than 60 years. The classification of the tumors according to the TNM (tumor, node, metastasis) staging system was as follows: 267 (50.76%) cases with T1 stage and 259 (49.24%) cases with T2-4 stage; 238 (45.25%) cases with N0 stage and 288 (54.75%) cases with N1/X stage; 418 (79.47%) cases with M0 stage, 106 (20.15%) cases with N1/X stage, and 2 (0.38%) cases of unknown stage; 239 (45.44%) cases with G1/2 stage, 279 (53.04%) cases with G3/4, and 8 (1.52%) cases of unknown stage. The classification of the tumors according to the Overall Stage Grouping (OSG) was as follows: 318 (60.46%) cases with stage I/II, 205 (38.97%) cases with stage III/IV, and 3 (0.57%) cases of unknown stage; 330 (60.46%) cases with stage I/II, 147 (38.97%) cases with stage III/IV, and 49 (0.57%) cases of unknown stage. The GSE40435 (101 normal subjects and 101 primary tumors), and GSE53757 (70 normal subjects and 70 primary tumors) data sets from the GEO database were used for verification.

## NEDD4L is downregulated in ccRCC

We screened the transcription levels of NEDD4 family members in the TCGA_KIRC. The differentially expressed genes (DEGs) between normal and tumor tissues were analyzed using the following criteria: fold change (FC) >2 and FDR< 0.05. The results revealed that only *NEDD4L* was meaningfully down-regulated in tumors compared with that in normal tissues (Fig. 1A). Subsequently, two data sets from the GEO database were used for verification, and the results confirmed that *NEDD4L* was significantly down-regulated in ccRCC carcinoma tissues (Figs. 1B–1C). Data from the Oncomine database made up of five separate queues also confirmed that *NEDD4L* was meaningfully down-regulated in ccRCC (Fig. 1D). Immunohistochemical results showed that the expression level of NEDD4L in renal cell carcinoma was significantly decreased than that in normal kindey (Fig. S1).

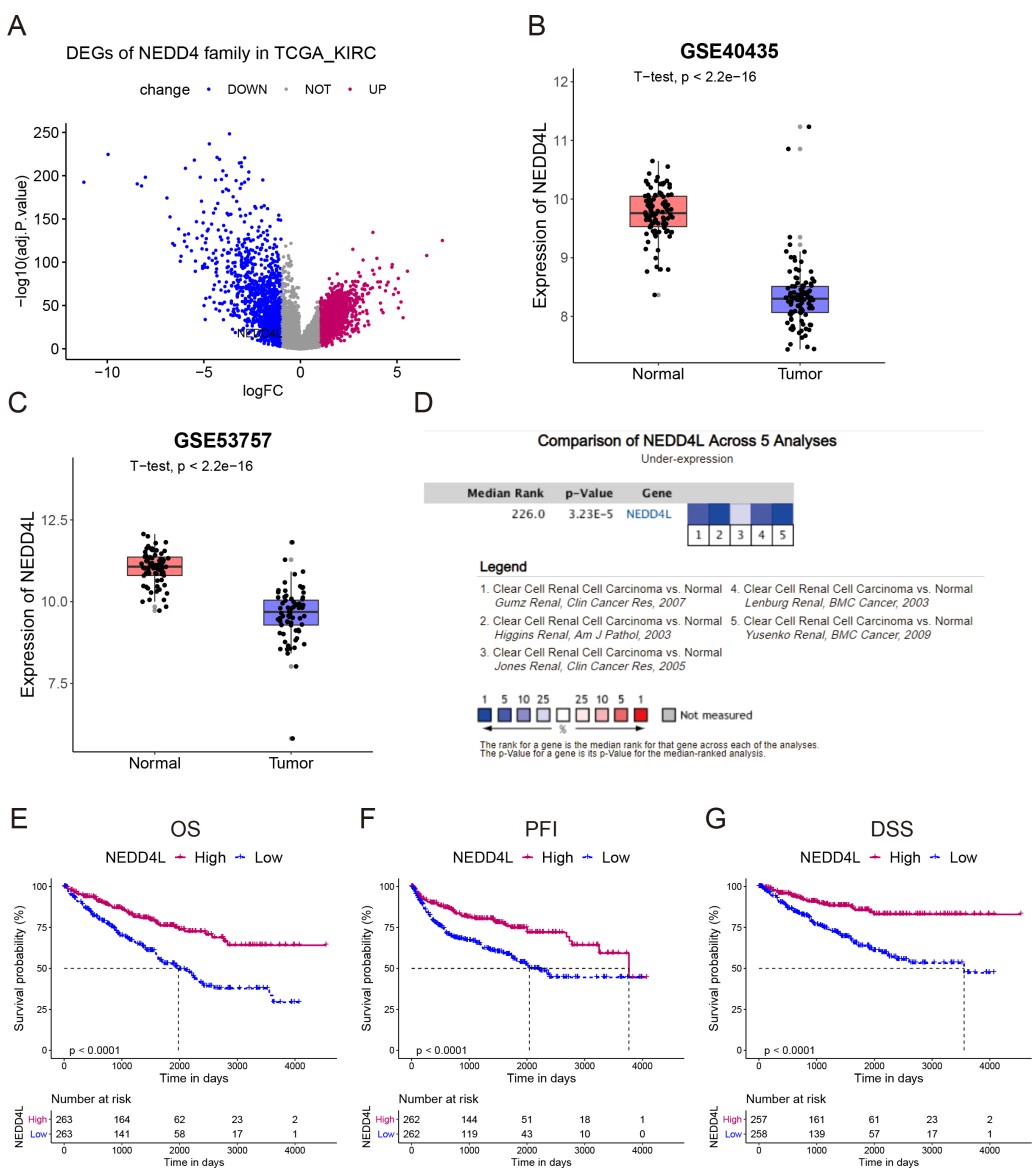

**Figure 1** **Expression and prognostic analysis of NEDD4L in ccRCC.** (A) In TCGA_KIRC data set, DEGs were represented in a volcano plot. (B & C) Mining of the GEO database showed that NEDD4L was downregulated in tumor tissues. (D) Data from the Oncomine database confirmed that NEDD4L was significantly downregulated in tumor tissues. (E) Kaplan–Meier curves of the OS of the patients with ccRCC. (F) Kaplan–Meier curves of the PFI of the patients with ccRCC. (G) Kaplan–Meier curves of the DSS of the patients with ccRCC.

## Low expression of NEDD4L correlated to dismal prognosis in ccRCC

The TCGA_KIRC data set was used to investigate the prognostic value of NEDD4L in ccRCC. After data filtering, a total of 526 samples with gene expression and survival data were included. The median of the gene expression was set as the cut-off value. Kaplan–Meier survival analysis showed that patients with low NEDD4L expression had shorter OS ($P < 0.0001$), PFS ($P < 0.0001$) and DSS ($P < 0.0001$) (Figs. 1E–1G). Patients

were divided into different subgroups according to sex, age, laterality, cancer status, histological grade, TNM stage, and OSG tumor stage. Analysis results of all but the N1 and M1 subgroups showed that patients with high *NEDD4L* expression had longer OS (Figs. 2 and 3, Figs. S2 & S3). Univariate and multivariate analysis were used to validate the prognostic value of NEDD4L in ccRCC. The univariate analysis results demonstated that low *NEDD4L* expression was associated with shorter OS (hazard ratio [HR]: 2.323; 95% CI [1.683−3.208]; $P = 0.001$) (Table 1). Multivariate COX regression analysis revealed that low expression of NEDD4L was still an independent factor associated with the deterioration of OS (HR: 1.905; CI [1.33–2.73], $P < 0.001$) (Table 1). These results suggested that NEDD4L may be an independent prognostic factor for ccRCC.

### NEDD4L downregulation correlated with worse clinicopathological features of ccRCC

To investigate the relationship between the expression of NEDD4L and clinicopathological features in renal cell carcinoma, patients were divided into low and high expression subgroup on the base of the median expression of NEDD4L. Chi-square test showed that lower levels of NEDD4L were significantly associated with adverse prognostic features, including ccRCC with higher T stages (OR = 0.456 (0.322−0.646) for T1 vs. T2/3/4, $P < 0.001$), M stage (OR = 0.530 (0.342−0.821) for M0 vs. M1/X, $P = 0.004$), OSG tumor stage (OR = 0.429 (0.299−0.615) for stage I/II vs. stage III/IV, $P < 0.001$), histological grade (OR = 0.456 (0.321−0.647) for G1/2 vs. G3/4, $P < 0.001$) (Table 2).

### Co-expression and enrichment analysis

Since NEDD4L was downregulated in ccRCC and correlated with worse prognosis of patients with ccRCC, we undertook to elucidate its potential role in this disease. In TCGA_KIRC and GSE40435 data sets, the correlation between the expression of NEDD4L and other genes in tumor samples was analyzed. A total of 494 genes in the TCGA_KIRC and 515 genes in the GSE40435 were identified as NEDD4L-related genes, using the following criteria: coefficient >0.5 and *p* value < 0.05. Among them, 206 NEDD4L-related genes were shared between the two datasets (Fig. S4, Table S2 ) and were subjected to GO and KEGG analysis. GO analysis revealed that the proteins encoded by these genes were mainly located in mitochondrial membrane and matrix, involved in the catabolism of many metabolites, such as lipid oxidation, fatty acid oxidation, fatty acid catabolism, monocarboxylic acid catabolism, and performed various molecular functions, such as DNA binding and ATP enzyme activity (Fig. 4A, Table S3). Accordingly, KEGG analysis showed that these genes play a major role in amino acid metabolism, oxidative phosphorylation and citrate cycle (Fig. 4B). These results suggest that NEDD4L is closely related to ccRCC cell metabolism.

### Development of a NEDD4L-related prognostic model

To develop a NEDD4L-related prognostic model for ccRCC, we first performed univariate Cox regression analysis on the aforementioned 206 NEDD4L-related genes, which identified 183 genes with significant prognostic relevance ($p < 0.05$, Table S2). Patients from TCGA_KIRC ($n = 526$) were randomly and equally divided into two groups, one is to

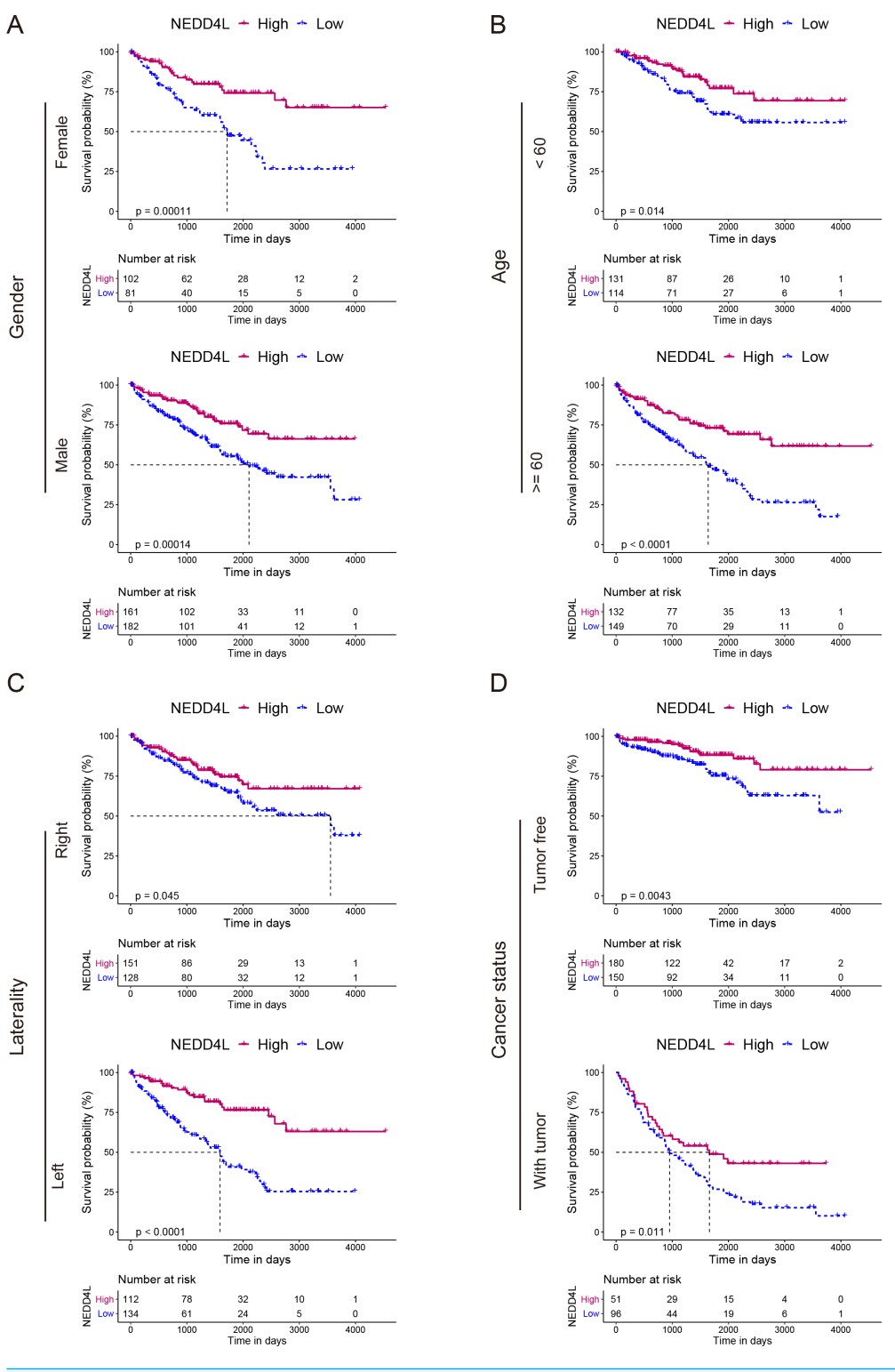

**Figure 2 Prognostic analysis of NEDD4L in ccRCC patients grouped by sex, age, laterality, and cancer status.** (A–D) Kaplan–Meier plot of the OS in high- and low-NEDD4L expression subgroups of ccRCC patients who were grouped according to sex (A), age (B), laterality (C), and cancer status (D).

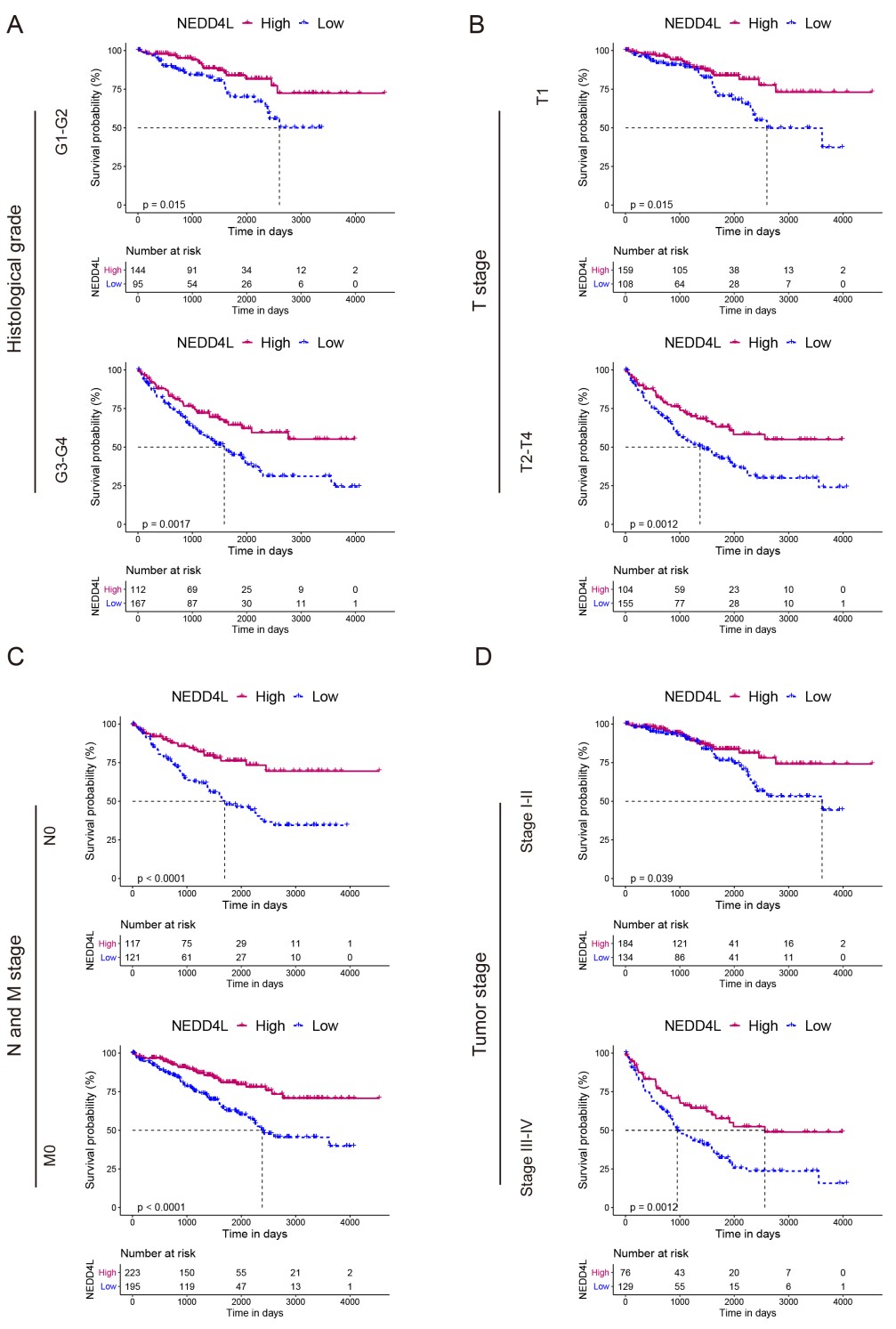

**Figure 3  Prognostic analysis of NEDD4L in ccRCC patients grouped by histological grade, TNM stage, and tumor stage.** (A–D) Kaplan–Meier plot of the OS in high- and low-NEDD4L expression subgroups of ccRCC patients who were grouped according to histological grade (A), TNM stage (B & C), and tumor stage (D).

**Table 1 Univariate and multivariate analysis of overall survival in TCGA patients with ccRCC.**

| Variables | Univariate analyses | | Multivariate analyses | |
|---|---|---|---|---|
| | Hazard ratio (95% CI) | *p*-value | Hazard ratio (95% CI) | *p*-value |
| Age (<60 *vs.* ≥60) | 0.557 (0.406–0.746) | <0.001 | 0.681 (0.487–0.953) | 0.025 |
| Sex (female *vs.* male) | 1.056 (0.773–1.442) | 0.734 | | |
| Laterality (left *vs.* right) | 1.409 (1.042–1.904) | 0.026 | 1.345 (0.972–1.860) | 0.073 |
| NEDD4L (low *vs.* high) | 2.323 (1.683–3.208) | <0.001 | 1.905 (1.330–2.730) | <0.001 |
| T (T1 *vs.* T2/3/4) | 0.336 (0.240–0.470) | <0.001 | 1.328 (0.707–2.495) | 0.378 |
| N (N0 *vs.*N1/X) | 1.106 (0.819–1.493) | 0.511 | | |
| M (M0 *vs.* M1/X) | 0.265 (0.195–0.362) | <0.001 | 0.828 (0.558–1.231) | 0.351 |
| Cancer status (tumor free *vs.* with tumor) | 0.195 (0.140–0.273) | <0.001 | 0.334 (0.225–0.495) | <0.001 |
| Stage (stage I vs. stage II/III/IV) | 0.253 (0.183–0.349) | <0.001 | 0.447 (0.240–0.834) | 0.011 |
| Grade (G1/2 *vs.* G3/4) | 0.374 (0.265–0.529) | <0.001 | 0.759 (0.508–1.134) | 0.178 |

**Table 2 *NEDD4L* expression associated with clinical characteristics.**

| Clinical characteristics | Total (N) | OR (95% CI) | *P* |
|---|---|---|---|
| T (T1 *vs.* T2/3/4) | 526 | 0.456 (0.322–0.646) | <0.001 |
| M (M0 *vs.* M1/X) | 524 | 0.530 (0.342–0.821) | 0.004 |
| N (N0 vs.N1/X) | 526 | 1.063 (0.754–1.499) | 0.726 |
| Stage (stage I/II *vs.* stage III/IV) | 523 | 0.429 (0.299–0.615) | <0.001 |
| Grade (G1/2 *vs.* G3/4) | 523 | 0.456 (0.321–0.647) | <0.001 |
| Sex (female *vs.* male) | 526 | 0.702 (0.490–1.008) | 0.055 |
| Age (<60 *vs.* ≥60) | 526 | 0.771 (0.547–1.087) | 0.137 |
| Laterality (left *vs.* right) | 525 | 1.411 (1.000–1.991) | 0.049 |
| Cancer status (tumor free vs. with tumor) | 477 | 0.443 (0.296–0.662) | <0.001 |

**Notes.**
OR, odds ratio; CI, confidence interval.
*P* values were calculated from two-sided chi-square test.

establish a LASSO Cox regression model, and the other is to evaluate the effectiveness of the model. The 183 NEDD4L-related genes with significant prognostic relevance were input into a LASSO Cox regression model, which generated eight key genes (Figs. 5A–5B), namely *DHRS7*, *NUPR2*, *C4orf19*, *CDKL2*, *SOWAHB*, *WDR72*, *EPB41L4A-DT*, and *IRF6* (Fig. 5C). The risk score was calculated by inputting the selected signature genes into the above formula. The median risk score was set to a cutoff value, dividing patients into low-risk subgroups and high-risk subgroups. Prognostic analysis with Kaplan–Meier method showed that patients with ccRCC in the low risk subgroup had significantly longer DSS (Figs. 6A–6C). The analysis of the time-related receiver operating characteristic (ROC) in the TCGA training data set shows that the risk score had a favorable predictive value (Fig. 6D, area under the curve (AUC) at 1 year = 0.72, AUC at 3 years = 0.75, AUC at 5 years = 0.8). The predictions of the established risk scores were further confirmed by the TCGA test data set and the entire TCGA data set. The ROC analysis of the TCGA testing dataset showed that the AUC was 0.76 at 1 year, 0.69 at 3 years, and 0.68 at 5 years (Fig. 6E), while that of the TCGA entire dataset showed that the AUC was 0.74 at 1 year, 0.71 at 3

A

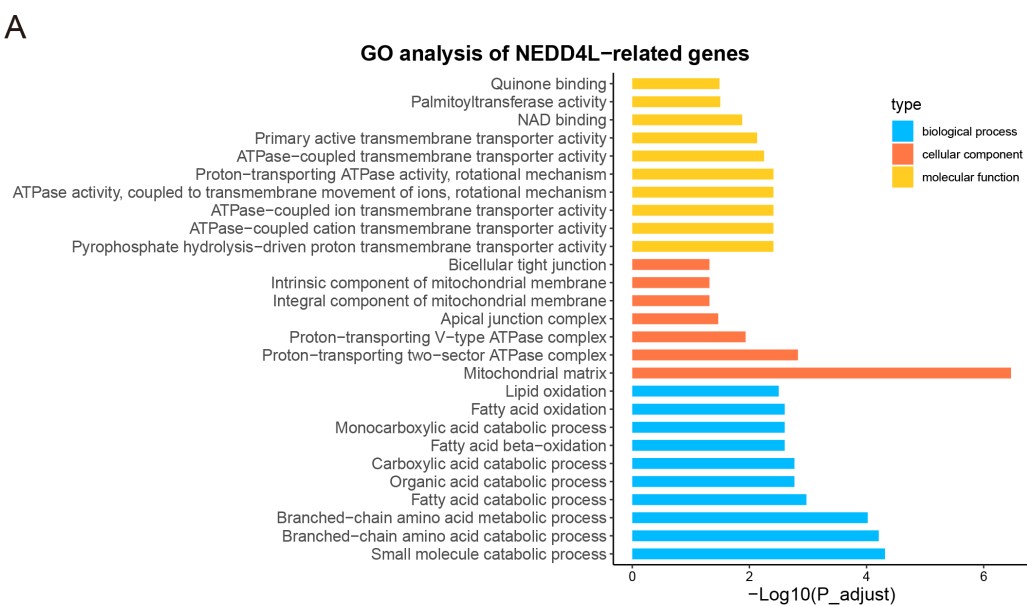

B

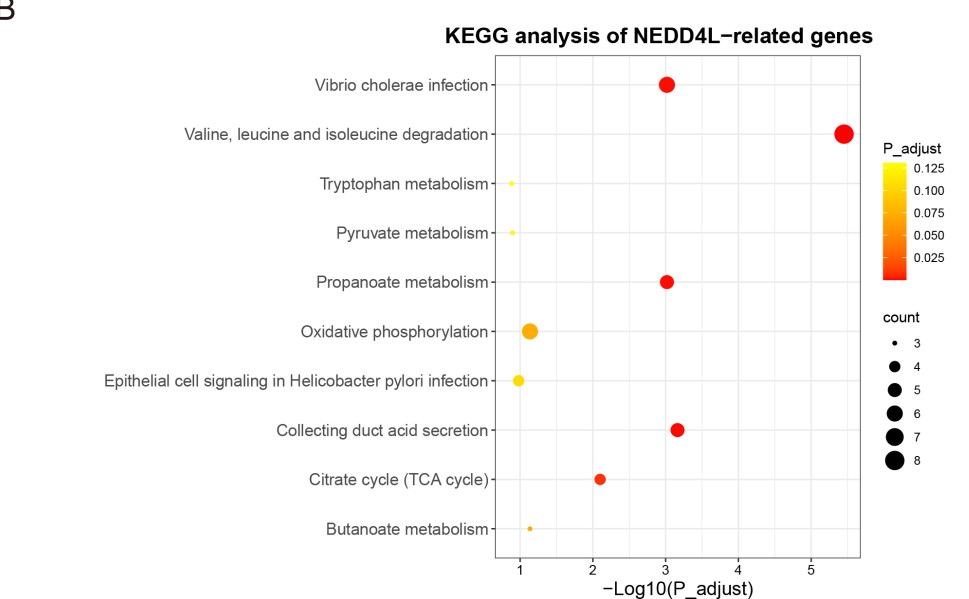

**Figure 4** **GO and KEGG enrichment analysis of NEDD4L.** (A) GO enrichment analysis of NEDD4L. (B) KEGG enrichment analysis of NEDD4L.

years, and 0.74 at 5 years (Fig. 6F). As reflected by the AUC values at 1 year, 3 year and 5 years, the signatures developed in our work had relatively better performance in predicting outcome of ccRCC patients (Fig. 6F, Fig. S5) (*Zhang, Qiu & Yang, 2021*; *Fei, Chen & Xu, 2021*; *Chu et al., 2021*; *Xing et al., 2021*).
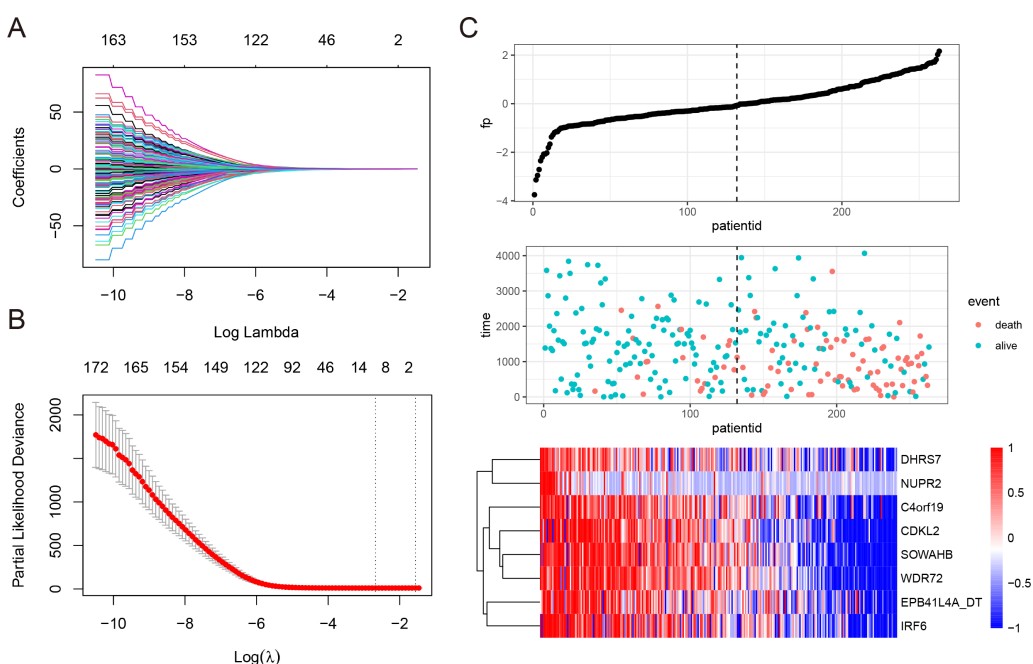

**Figure 5  Construction of LASSO Cox regression model.** (A–C) A LASSO Cox regression model was developed from NEDD4L and 183 related genes, calculating the tuning parameter ($\lambda$) based on the partial likelihood deviance with tenfold cross-validation. Each dot represents a patient, and the heatmap represents the genes expression levels. The optimal log $\lambda$ value is indicated by the vertical black line in the plot.

## Identification of NEDD4L-related signaling pathways by GSEA

Various biological processes are recognized to be involved in the occurrence and development of ccRCC. Accordingly, we hypothesized that the worse survival related to low expression of NEDD4L may be associated with some activated signal pathways in ccRCC. To evaluate this hypothesis, we performed a GSEA of the differences in the NEDD4L-related low risk group in TCGA_KIRC to verify the key signaling pathways related to NEDD4L expression. There were significant differences in the richness of MSigDB sets (c2.cp.v6.2 symbols) of multiple pathways (normalized $P < 0.05$, FDR $< 0.25$). According to the normalized enrichment score (NES), the signaling pathways most significantly associated with NEDD4L expression are shown in Fig. 7 and Table 3. In particular, NEDD4L expression was associated with biosynthesis of unsaturated fatty acids, insulin signaling pathway, and fatty acid metabolism, supporting a possible role of NEDD4L in the regulation of various metabolic processes

## Establishment of a nomogram based on NEDD4L

Univariate and multivariate Cox regression analysis were achieved on the TCGA_KIRC ccRCC data set to investigate whether the risk score was an independent predictor of prognosis in patients with ccRCC. The adjustment of conventional clinical models, including age, sex, laterality, TNM stage, cancer status, histological grade, and tumor stage, indicated that the risk score was also an independent prognostic factor, which confirmed

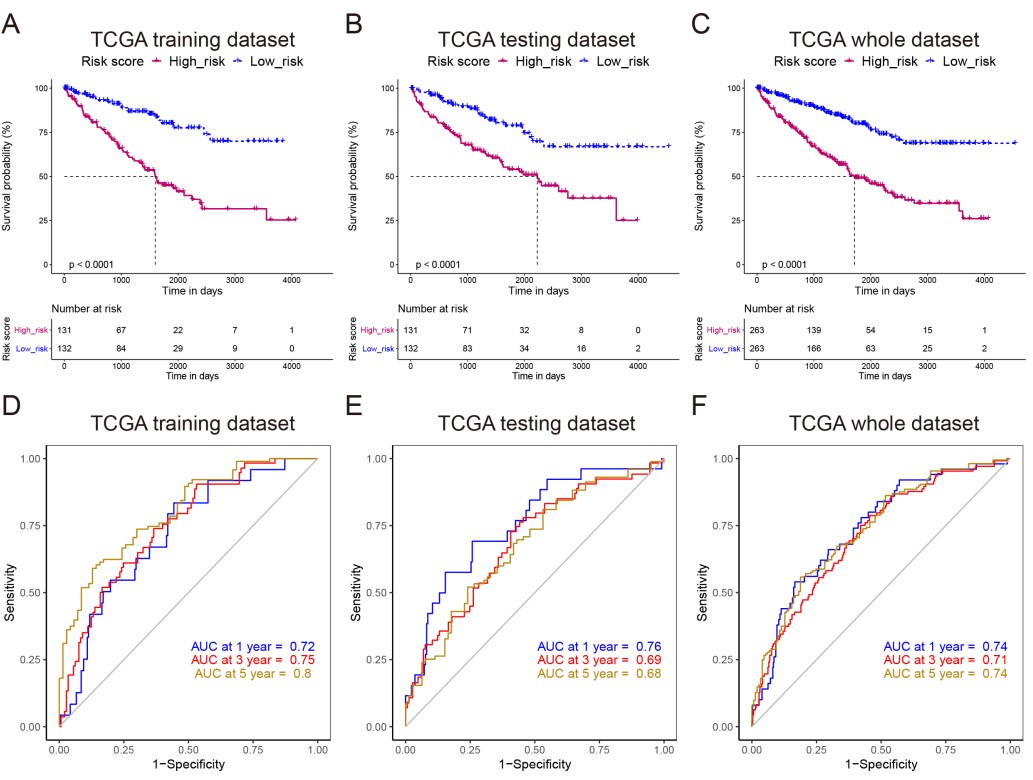

**Figure 6 Reliability verification of NEDD4L-related prognostic model.** (A–C) Kaplan–Meier plot of the DDS in the high- and low-risk groups of the TCGA_KIRC cohort. (D–F) Time-dependent ROC analysis of the risk score for the DDS and survival status in the TCGA_KIRC cohort.

it had a strong ability to predict OS in patients with ccRCC (OR = 1.9329 (1.3429–2.782), $P < 0.001$, Figs. 8A–8B)

To establish a quantitative method for ccRCC prognosis, we used independent clinical risk factors to develop a nomogram (Fig. 8C). The point scale of nomogram is used to assign points to each variable based on multivariate Cox analysis. We drew a horizontal line to determine the points of each variable, then added the points of all variables together to calculate the total score of each patient, and then normalize it to a distribution of 0 to 45. The estimated 1 -, 3-and 5-year survival rates of ccRCC patients were calculated by drawing a vertical line between the total point axis and each prognostic axis (*Li et al., 2019a*; *Li et al., 2019b*). The C-index of the nomogram 0.7627102 (0.7255857–0.7998348) suggesting a good prediction effect. Thus, our results prove that nomogram was the best model for predicting the prognosis of renal cell carcinoma compared with a single risk factor.

## DISCUSSION

The Nedd4-like E3 family includes nine members, which regulate key signaling pathways in tumorigenesis, such as TGFβ, EGF, IGF, VEGF, CXCL12, and TNF (*Chen & Matesic, 2007*). In cancers, NEDD4, WWP1, WWP2, SMURF1, and SMURF2 have been shown to function as oncogenes, while NEDD4L acts as an antioncogene (*Chen & Matesic, 2007*;

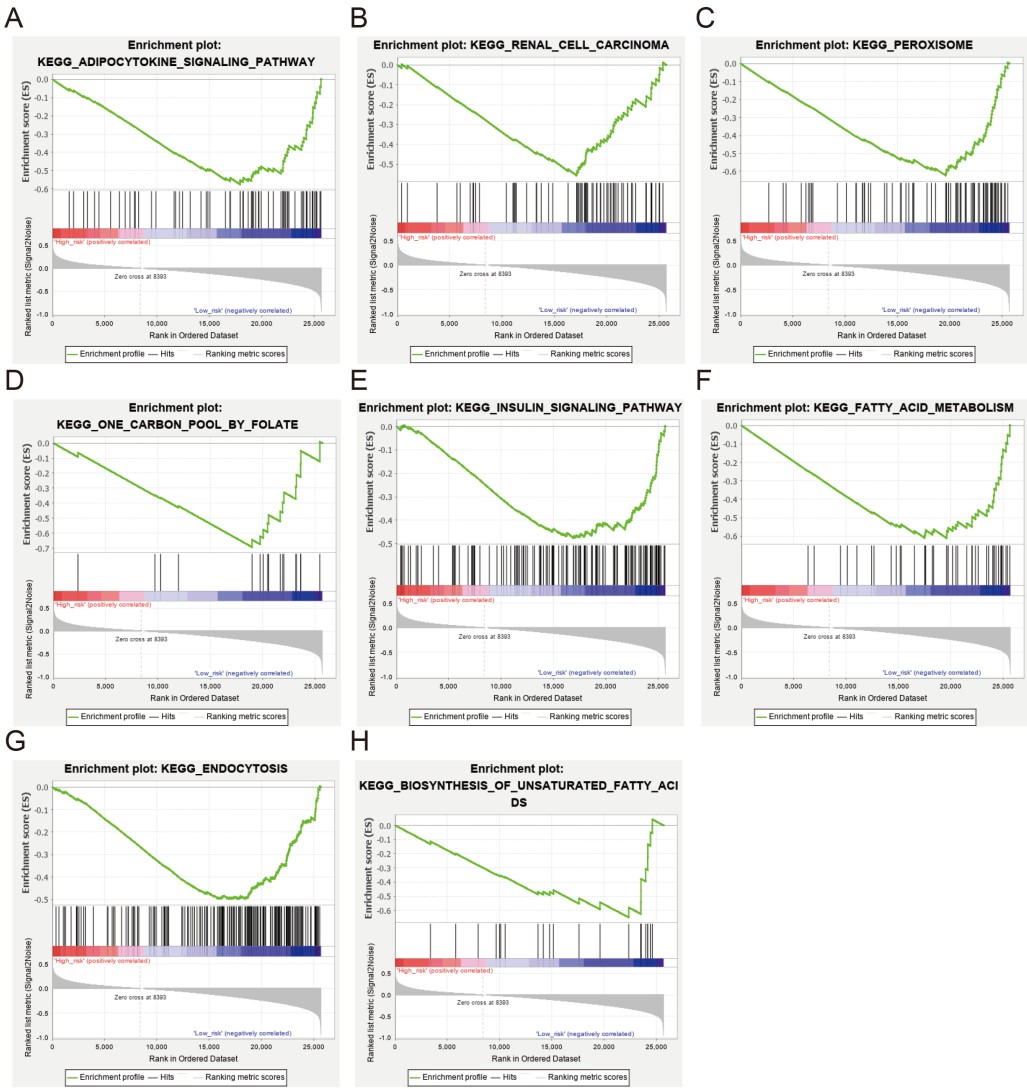

**Figure 7   GSEA between ccRCC patients with high and low NEDD4L expression in the TCGA_SKCM and GSE40435 cohorts (A–H).**

*Tanksley, Chen & Coffey, 2013*; *Zhao et al., 2015*). This study systematically investigated the role of the NEDD4 family in ccRCC through the comprehensive analysis of TCGA_KIRC and GEO data sets and Oncomine database. The analysis revealed that only NEDD4L was significantly downregulated in ccRCC, and low expression of NEDD4L was associated with shorter survival (Fig. 1).

NEDD4 is shown to be overexpressed in various types of human cancers including gastric cancer (*Kim et al., 2008*; *Sun et al., 2014*; *Reichert-Penetrat et al., 1998*), colorectal cancer (*Kim et al., 2008*), breast cancer (*Kim et al., 2008*), non-small-cell lung carcinoma (*Amodio et al., 2010*; *Shao et al., 2018*), and hepatocellular carcinoma (*Hang et al., 2016*; *Huang et al., 2017*). NEDD4 acts as a double-edged sword in tumors. On the one hand, NEDD4 promotes tumor survival by inducing cell proliferation, inhibiting apoptosis,

**Table 3   GSEA of the NEDD4L-related low risk group in TCGA_KIRC.**

| MSigDB collection | Gene set name | NES | NOM *p*-value |
|---|---|---|---|
| c2.cp.v6.2.symbols.gmt | ERBB signaling pathway | 1.812 | 0.002 |
| | Vasopressin regulated water reabsorption | 1.713 | 0.012 |
| | Propanoate metabolism | 1.708 | 0.041 |
| | Neurotrophin signaling pathway | 1.660 | 0.008 |
| | Ubiquitin mediated proteolysis | 1.653 | 0.018 |
| | RNA degradation | 1.650 | 0.008 |

**Notes.**
NES, normalized enrichment score; NOM, nominal; FDR, false discovery rate.
Gene sets with NOM *p*-value < 0.05 and FDR *q*-value < 0.25 are considered as significant.

disrupting the cell cycle, promoting cell migration and invasion, and enhancing drug resistance (*Wang et al., 2020*). On the other hand, NEDD4 inhibits tumor proliferation by binding to MYC, inhibiting HER3 expression, and targeting PIP5K1A. SMURFs also plays a dual role in tumors, both as tumor promoters and suppressors (*David, Nair & Pillai, 2013*). By controlling the stability of several key proteins, these proteins play a central role in cell cycle progression, proliferation, differentiation, metastasis, genome stability and senescence. WWP1 is frequently upregulated in multiple human malignancies, and has been found to play an important role in cell proliferation, apoptosis and invasion (*Nourashrafeddin et al., 2015*; *Zhi & Chen, 2012*; *Chen et al., 2007a*; *Chen et al., 2007b*; *Chen & Zhang, 2018*). WWP2 functions as an oncogene in liver cancer, breast cancer, endometrial cancer, gastric cancer, lung cancer, oral cancer, and ovarian cancer (*Clements et al., 2015*; *Fang et al., 2020*; *Fukumoto et al., 2014*; *Jung et al., 2014*; *Wang et al., 2020*; *Butt et al., 2019*; *Sakashita et al., 2013*; *Wan et al., 2019*). The expression of NEDL1 in benign neuroblastoma is higher than that in non-malignant neuroblastoma (*Li et al., 2008a*; *Li et al., 2008b*). So far as we know, there are no reportes on ITCH and NEDL2 in tumors, thus their function in ccRCC is unclear. Our study showed that their expression levels were no significantly changed in ccRCC, which is inconsistent with reports in other tumors, which may be due to tumor specificity.

NEDD4L has been considered as a tumor suppressor. Decreased NEDD4L expression has been found in colorectal cancer (*Tanksley, Chen & Coffey, 2013*), gastric cancer (*Jiang et al., 2019*), ovarian cancer (*Yang et al., 2015*), prostate cancer (*Hu et al., 2009*), and non-small cell lung cancer (*Hu et al., 2009*). Consistent with previous reports, this study suggested that NEDD4L acts as a tumor suppressor in ccRCC. Subsequent analysis showed that NEDD4L is an independent prognostic factor for ccRCC and might be associated with energy metabolism. Lee et al. reported that NEDD4L inhibited autophagy and mitochondrial metabolism by downregulating the expression of ULK1 or ASCT2, thereby inhibiting the growth and survival of pancreatic cancer cells. *Kim et al. (2018)* reported that NEDD4L acts a pivotal part in the feedback regulation of cAMP signal by limiting CREB-regulated transcription coactivator 3 protein levels. The findings of the above studies are consistent with the findings of our study, indicating that NEDD4L plays an important role in tumor metabolism.

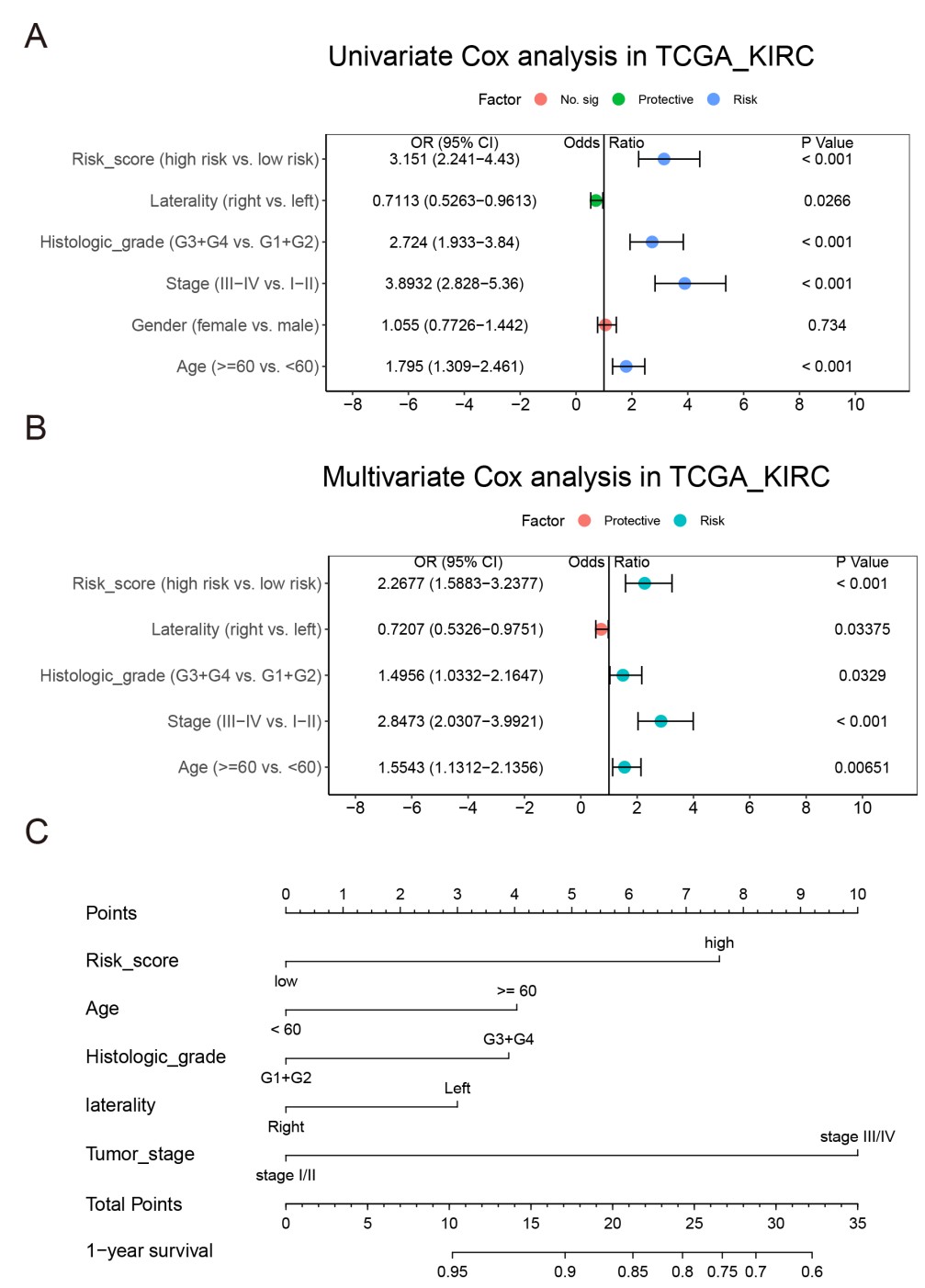

**Figure 8  Integration of risk score and clinical characteristics.** (A & B) Univariate and multivariate regression analysis of the relation between the risk score and clinicopathological characteristics regarding OS in the TCGA_KIRC. (C) Nomogram developed to predict the 1-, 3-, and 5-year OS in the TCGA_KIRC data set.

We also established a prognostic model based on NEDD4L using the LASSO Cox regression, a broadly selected machine learning algorithm used to minimize the risk of overfitting. The model had good predictive value for training and test queues (Fig. 6). A total of 526 cases from TCGA_KIRC were randomly divided into low-risk and high-risk subgroups, and then analyzed by GSEA, GO, and KEGG enrichment analysis, which are widely used bioinformatics tools in the functional characterization of specific genes.

All of these enrichment analyses suggested that the main difference between the two groups is in metabolism (Figs. 4 and 7, Table 3). The decreased expression of NEDD4L may lead to metabolic disorder and promote the development of ccRCC.

Lee et al. (2020) study found that NEDD4L could inhibit the growth and survival of pancreatic cancer cells by inhibiting autophagy and mitochondrial metabolism. Zhang et al. reported that NEDD4L also played vital roles in bone metabolism. These studies provided strong evidence for the involvement of NEDD4L in cell metabolism (PMID: 22957059). The prediction model developed in this study showed that DHRS7, NUPR2, C4orf19, CDKL2, SOWAHB, WDR72, EPB41L4A_DT, and IRF6 were significantly associated with NEDD4L function.

DHRS7 can recognize steroids and retinoids as potential substrates, and might act a pivotal part in the metabolism of these signal molecules (Stambergova et al., 2016). Lopez et al. (2015) reported that NUPR2 could induce G1 phase cell cycle arrest, decrease cell survival rate and proliferation ability. C4orf19 was found to be significantly downregulated in multicentric breast cancer, but its function is unclear (Lang et al., 2017). Two studies showed that the decreased expression of CDKL2 was associated with the progression and poor prognosis of gliomas (Yi et al., 2018; Yi et al., 2020), and another study showed that CDKL2 promoted EMT and progression of breast cancer (Li et al., 2014). WDR72 was found to be crucial in ameloblasts and the kidneys, and may act a pivotal part in enamel maturation (Rungroj et al., 2018). EPB41L4A_DT was reported to inhibit the progression of ovarian cancer, hepatocellular carcinoma, non-small cell lung cancer, and breast cancer (Sun, Yang & Gao, 2020; Wang et al., 2019; Shu et al., 2018; Xu et al., 2016). IRF6 has been proved to be a tumor suppressor in breast and gastric cancer (Li et al., 2019a; Li et al., 2019b; Xu et al., 2019). All these genes play important roles in a variety of solid tumors, indirectly confirming the importance of NEDD4L in ccRCC.

A NEDD4L-related prognostic model of ccRCC was established by LASSO Cox regression. After adjusting the routine clinical features, the risk score was confirmed as an independent prognostic factor. It seemed that the risk score may have a stronger predictive power than the traditional prognostic factors. We also conducted a comprehensive assessment in combination with risk scores and other important clinical models. According to the calibration graph, there was a significant consistency between the actual and predicted values of the 1 -, 3-and 5-year OS (Fig. 8). In order to show the reliability of our risk model, we compared it with other ccRCC risk models, and the results showed that our risk model has better prediction ability (Fig. S5) (Zhang, Qiu & Yang, 2021; Fei, Chen & Xu, 2021; Chu et al., 2021; Xing et al., 2021). Based on the complementary perspective of the corresponding tumor, the model provides a personalized score for individual patients and a valuable new prognostic evaluation method for clinicians.

Although this study provides new clues for the relationship between NEDD4L expression and ccRCC progression, it still has some limitations. First, it is necessary to further verify the function of NEDD4L in ccRCC and cellular metabolic processes, and completely elucidate the specific mechanism involved. Second, our risk score requires a large sample of prospective studies to be demonstrated, which is just a retrospective analysis.

## ACKNOWLEDGEMENTS

We thank Editorbar for providing language assistance in the preparation of this article.

### Funding

This work was supported by the National Natural Science Foundation of China (81903031), China Postdoctoral Science Foundation (2020M682334), Henan Postdoctoral Foundation (202003002), the Open Project of State Key Laboratory of Cancer Biology of China (CBSKL2019KF12, CBSKL2019KF13). The funders had no role in study design, data collection and analysis, decision to publish, or preparation of the manuscript.

### Grant Disclosures

The following grant information was disclosed by the authors:
National Natural Science Foundation of China: 81903031.
China Postdoctoral Science Foundation: 2020M682334.
Henan Postdoctoral Foundation: 202003002.
Project of State Key Laboratory of Cancer Biology of China: CBSKL2019KF12, CBSKL2019KF13.

### Competing Interests

The authors declare there are no competing interests.

### Author Contributions

- Hui Zhao, Junjun Zhang and Xiaoliang Fu performed the experiments, prepared figures and/or tables, authored or reviewed drafts of the paper, and approved the final draft.
- Dongdong Mao, Xuesen Qi, Gang Meng, Zewen Song, Ru Yang, Zhenni Guo, Binghua Tong and Meiqing Sun analyzed the data, prepared figures and/or tables, and approved the final draft.
- Baile Zuo and Guoyin Li conceived and designed the experiments, authored or reviewed drafts of the paper, and approved the final draft.

### Data Availability

The raw data are available from TCGA (project ID TCGA_KIRC) and from NCBI GSE40435 and GSE53757.

## Supplemental Information

Supplemental information for this article can be found online at http://dx.doi.org/10.7717/peerj.11880#supplemental-information.

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
