# Peer review of "Integrated bioinformatics analysis of the NEDD4 family reveals a prognostic value of NEDD4L in clear-cell renal cell cancer"

_PeerJ, doi:10.7717/peerj.11880_

## Round 0.1 · original submission · Major Revisions

Please address all critiques of both reviewers and revise your manuscript accordingly.

Reviewer 1 ·

Basic reporting

no comment

Experimental design

There are thousands of differentially expressed genes in cancers as compared to normal tissue. The authors did not show how NEDD4L-related genes are important in this background. The authors could develop a prognostic model that was derived from NEDD4L-related genes, but it can be speculated that this prognostic model is not the best one (although the ROCs with AUC seem to be good). A good prognostic model should be generated from the genes screened by survival analysis of all candidates. The authors did not show the advantage of NEDD4L-based prognostic model over other models primarily aimed to predict prognosis, and this would limit the use and follow of this model.

Validity of the findings

no comment

Additional comments

This study conducted an integrated bioinformatics analysis using public datasets and identified NEDD4L as the differentially expressed gene in ccRCC compared with normal samples. The gene this study identified may have potential role in tumorigenesis and values of diagnosis and target treatment for ccRCC. This study suffers from some limitations that need to fully addressed.

Major concerns:
There are thousands of differentially expressed genes in cancers as compared to normal tissue. The authors did not show how NEDD4L-related genes are important in this background. The authors could develop a prognostic model that was derived from NEDD4L-related genes, but it can be speculated that this prognostic model is not the best one (although the ROCs with AUC seem to be good). A good prognostic model should be generated from the genes screened by survival analysis of all candidates. The authors did not show the advantage of NEDD4L-based prognostic model over other models primarily aimed to predict prognosis, and this would limit the use and follow of this model.

It's not usual to divide the tumor stage into stage I vs. stage II-IV in survival analysis. Stage I-II vs III-IV may be a better choice.

Prognostic value of dichotimized NEDD4L seemed to be tested in a single cohort (TCGA). How the cutoff of NEDD4L was determined? The prognostic value needs to be validated in other cohorts with public survival information and gene expression data.

The authors discovered NEDD4L as a critical gene in ccRCC. However, they should validate it in the author's in-house cohort and tumor samples.

Figure 8C, the workflow and method for generating nomogram are not reliable. In the nomogram, both M-stage and TNM stage are included in the model. However, M stage is a part of TNM staging system. It's werid to see 'Tumor-free' and 'with tumor' in the nomogram. How these two end-point status could be selected as predictive variables for survival?

line 105, "The correlation between NEDD4L and fatty acids has rarely been studied. One study showed that 20‑hydroxyeicosatetraenoic acid regulates NEDD4L expression in kidney and liver[30]." GSEA analysis support this one. However, the authors did not show how their GO, KEGG, and GSEA results could support the other popular roles of NEDD4L in ccRCC. This is important, since it could verify the robust of authors' methodology in this manuscript.

The authors defined NEDD4L-related genes using correlation analysis. However, interaction network analysis would be more robust, as they seemed to find a set of genes regulated or impacted by NEDD4L and take them to downstream enrichment analysis.



Minor concerns:
More details on quantification of gene expression should be provided. Is RNA-seq or microarray was used in the analysis?
Table 3 and Figure 7 need to be edited, the terms should not be presented as a computer input way. The p values and enrich scores should be shown in each panel of figure 7.
line 221, q value or adjusted p values in multiple correlation test should be shown there.
Legend of figure 5 did not describe the panel of heatmap and dotter plot on the right bottom.
The bulk survival plots in figure 2-3 need to be edited and cut.

Reviewer 2 ·

Basic reporting

This study proposed a bioinformatics analysis to understand the role of NEDD4 family in clear-cell renal cell cancer. The authors finally convinced that this gene may be a prognostic biomarker for ccRCC. The idea is reasonable; the use of clear and unambiguous English. There are some major points that need to be addressed:

- Literature review is weak. There must have some related works on bioinformatics analysis on CcRCC.

- Quality of figures should be improved.

- "Abstract" should not be broken into two paragraphs.

- There should have space before reference number.

Experimental design

- It is unclear how did the authors divide/separate their datasets. How did they use GEO datasets since they only mentioned that analyzed 526 tumor samples from the survival data of the TCGA_KIRC data set.

- Methodology has not been explained clearly.

- Number of control samples is much lower than patients.

- Did the authors concern about the batch effect removal among the datasets?

- ROC curves have been used in previous bioinformatics studies i.e., PMID: 31750297, PMID: 31362508, and PMID: 33260643. Therefore, the authors are suggested to refer to more works in this description.

Validity of the findings

- The authors should show p-values on GO analysis.

- According to the ROC curves and AUC values, the performance results were not too high.

- The authors should compare the predictive performance or the finding biomarkers to previous works on the same problem/data.

Additional comments

No comment.

---

## Round 0.2 · Major Revisions

Although one of the reviewers was satisfied by your revision, another reviewer continues to believe that you should work closely with the statisticians and data scientists to check your data analysis, re-analyze the dataset and revise the whole text to present reliable results. I agree with these suggestions and ask you to follow these and other recommendations of reviewer #1.

Reviewer 1 ·

Basic reporting

NA

Experimental design

NA

Validity of the findings

NA

Additional comments

I still have multiple major concerns that may impact the reliability of this manuscript:

1. As the author declaimed in the revised Methods, they used HTSeq-FPKM. Unfortunately, this reviewer highly suggest HTseq-counts as an optimal method to quantify the mRNA expression. They may check the difference between FPKM and counts in the differential expression analysis across samples. This reviewer highly suggest the authors work closely with statisticians and data scientists to check their data analysis, re-analyze the dataset and revise the whole text to present reliable results.

2. Why skin melanoma from TCGA-SKCM was used in figure 7?

3. The results and discussion need to be more relevant to their hypothesis that NEDD4L is assoicated with fatty acids. In GSEA, GO, and KEGG analysis, they need to illustrate how fatty acids-related genes and pathways were critical and outstanding among others in NEDD4L-related differentiation.

Reviewer 2 ·

Basic reporting

No comment.

Experimental design

No comment.

Validity of the findings

No comment.

Additional comments

My previous comments have been addressed well.

---

## Round 0.3 · accepted · Accept

Since all remaining concerns are addressed, the revised manuscript is acceptable now.

Reviewer 1 ·

Basic reporting

NA

Experimental design

NA

Validity of the findings

NA

Additional comments

I appreciate the authors' response. My concerns have been addressed.